# Simultaneous Increase in CO_2_ and Temperature Alters Wheat Growth and Aphid Performance Differently Depending on Virus Infection

**DOI:** 10.3390/insects11080459

**Published:** 2020-07-22

**Authors:** Ana Moreno-Delafuente, Elisa Viñuela, Alberto Fereres, Pilar Medina, Piotr Trębicki

**Affiliations:** 1Escuela Técnica Superior de Ingeniería Agronómica, Alimentaria y de Biosistemas, Universidad Politécnica de Madrid (ETSIAAB-UPM), Avd. Puerta de Hierro 2-4, 28040 Madrid, Spain; ana.moreno@upm.es (A.M.-D.); elisa.vinuela@upm.es (E.V.); pilar.medina@upm.es (P.M.); 2Agriculture Victoria Research, Department of Jobs, Precincts and Regions, 110 Natimuk Rd, Horsham, VIC 3400, Australia; 3Associate Unit IVAS (CSIC-UPM): Control of Insect Vectors of Viruses in Horticultural Sustainable Systems, 28006 Madrid, Spain; 4Instituto de Ciencias Agrarias, Consejo Superior de Investigaciones Científicas (ICA-CSIC), C/Serrano 115 dpdo., 28006 Madrid, Spain; a.fereres@csic.es

**Keywords:** aphids, *Barley yellow dwarf virus*, changing environment, climate change, elevated carbon dioxide, increased temperature, *Rhopalosiphum padi* L., *Triticum aestivum* L., vector-plant-pathogen interactions

## Abstract

Climate change impacts crop production, pest and disease pressure, yield stability, and, therefore, food security. In order to understand how climate and atmospheric change factors affect trophic interactions in agriculture, we evaluated the combined effect of elevated carbon dioxide (CO_2_) and temperature on the interactions among wheat (*Triticum aestivum* L.), *Barley yellow dwarf virus* species PAV (BYDV-PAV) and its vector, the bird cherry-oat aphid (*Rhopalosiphum padi* L.). Plant traits and aphid biological parameters were examined under two climate and atmospheric scenarios, current (ambient CO_2_ and temperature = 400 ppm and 20 °C), and future predicted (elevated CO_2_ and temperature = 800 ppm and 22 °C), on non-infected and BYDV-PAV-infected plants. Our results show that combined elevated CO_2_ and temperature increased plant growth, biomass, and carbon to nitrogen (C:N) ratio, which in turn significantly decreased aphid fecundity and development time. However, virus infection reduced chlorophyll content, biomass, wheat growth and C:N ratio, significantly increased *R. padi* fecundity and development time. Regardless of virus infection, aphid growth rates remained unchanged under simulated future conditions. Therefore, as *R. padi* is currently a principal pest in temperate cereal crops worldwide, mainly due to its role as a plant virus vector, it will likely continue to have significant economic importance. Furthermore, an earlier and more distinct virus symptomatology was highlighted under the future predicted scenario, with consequences on virus transmission, disease epidemiology and, thus, wheat yield and quality. These research findings emphasize the complexity of plant–vector–virus interactions expected under future climate and their implications for plant disease and pest incidence in food crops.

## 1. Introduction

The concentration of carbon dioxide (CO_2_) in the atmosphere has increased by 130 ppm since the Industrial Revolution, largely due to intensified fossil fuel emissions and deforestation. Remarkably, 70% of the increase occurred in the last 60 years, with current CO_2_ concentrations surpassing 400 ppm. By the end of the century, equivalent CO_2_ concentration will reach 720–1000 ppm, which is estimated to increase global surface temperature by 2.2 °C [1], further changing the earth’s climate and affecting biological functions and ecosystem stability [2,3].

Pest and disease pressure can increase as a consequence of climate change due to changes in their latitudinal and altitudinal distribution, spatial and temporal mismatches between pests and their natural enemies, winter survival, or weakening of crop defenses [4,5], therefore compromising food security [6,7,8]. Insect herbivores are affected by climate and atmospheric factors, principally temperature [9], but also CO_2_, ultraviolet radiation, or rainfall, which will influence their development, survival, range, and abundance [3,10]. The incidence and severity of plant viruses are also affected by climate change due to mediated effects on plants and their insect vectors [11,12,13,14,15,16,17].

Elevated CO_2_ (eCO_2_) is known to increase plant biomass and yield and to reduce stomatal conductance and transpiration, with consequent increases in water use efficiency, photosynthetic rates, and net primary productivity [18,19,20,21]. Elevated CO_2_ changes the carbon to nitrogen (C:N) ratio in plants, generally increasing C content and reducing N [14,22,23]. Apart from the decrease in the N quantity, eCO_2_ also modifies the composition of N compounds, i.e., N quality, such as amino acids and proteins [24,25]. Most of the effects of eCO_2_ on insects and pathogens are indirect, mediated by changes in plant growth and plant physiology and biochemistry [3,26,27]. Because insects have high N requirements [24], the reduction in plant N content due to eCO_2_ impacts their feeding, development, longevity, and behaviour [23,25,28,29,30]. Among sap-feeding insects, positive, negative, and no responses due to eCO_2_ have been reported [13,22,28,30,31,32,33,34,35]. For aphids, responses due to eCO_2_ can be species- [28] or even genotype-specific [23,36] and are also dependent on the interaction between aphid N requirements and soil N fertility [37]. Furthermore, eCO_2_ can intensify the incidence of plant viruses [11], increasing virus-infected plant biomass, photosynthesis rate, and water use efficiency [15], as well as virus replication, with extended implications in virus transmission by aphid vectors [38,39].

Elevated temperatures (eT) can also modify plant nutrients, defenses, growth, phenology and physiology. Moderate increase in temperature generally rises photosynthesis and respiration rates, however heat waves or extreme temperatures can inhibit photosynthesis, increase respiration and finally decrease plant growth rates [27]. When several climate variables interact simultaneously, as it is a true representation of future climate, different effects could be accentuated (synergism, antagonism, addition, etc.) and, in some cases, eT can counteract the positive effects of eCO_2_ on plants [2,40,41]. In contrast to eCO_2_, temperature has a direct impact on invertebrates [9] as they are ectothermic and sensitive to temperature change [2]. Increasing temperature could increase insect development rates, fecundity, and the number of generations [2,9], as long as temperature remains within the insect thermal range for development. However, extreme temperature events can alter temporal and spatial structure of natural communities, with changes in demographic rates and fitness of species [42]. Increasing temperature could affect virus incidence in some pathosystems, e.g., triggering earlier and greater barley yellow dwarf virus (BYDV) titre in wheat (*Triticum aestivum* L.), and earlier expression of virus symptoms [43].

To generate more realistic conclusions about how climate change affects multi-trophic interactions, research need to consider changes in different climate and atmospheric variables as a whole, as these factors will occur simultaneously in the future [2,41,44,45]. Singling out any particular variables of the future climate and applying then separately (either eCO_2_ or temperature) on a complex biological system can lead to false conclusions of the future interactions [17]. However, to date, most of the studies on climate change affecting plant-insect-pathogen interactions have been conducted under one single abiotic factor (mainly either CO_2_ or temperature), as examined above [13,14,46,47,48]. The limited number of experiments carried out combining eCO_2_ and eT on plant–herbivore interactions have demonstrated that the effects could be modified or even mitigated because of the interrelation of the two factors [2,40,41,49]. Fewer studies have focused on pathosystems under future CO_2_ and temperature conditions [16,44]. For instance, several species of aphids have reduced development time while increased fecundity under eCO_2_ and eT combined subsequently increased their rates of increase, therefore potentially increasing the number of generations under future climate scenarios [50], which could intensify virus transmission and dispersion. Furthermore, the combination of eCO_2_ and eT can increase the number of winged aphids, which might enhance aphid dispersal and, when viruliferous, the spread of plant viruses [35].

Barley yellow dwarf virus species PAV (BYDV-PAV) is a member of the genus Luteovirus (family Luteoviridae) that causes plant yellowing, stunting, delayed heading, and reduced yields in wheat, barley (*Hordeum vulgare* L.), and other members of the family Poaceae [51], with symptoms and severity of infection dependent on plant host, cultivar resistance, time of infection and climatic conditions [52,53,54,55]. The virus is phloem-limited and transmitted obligatorily by aphid vectors in a circulative, non-propagative manner [56,57]. BYDV-PAV is mainly transmitted by the bird cherry-oat aphid *Rhopalosiphum padi* (L.) (Hemiptera: Aphididae) [58], which has global distribution and is among the few aphid species considered to be one of the most economically important agricultural pests worldwide [59].

The principal aim of our research was to investigate the effects of simultaneous elevated CO_2_ and temperature on the interactions among wheat, BYDV-PAV, and *R. padi*, as no empirical studies have included these climate and atmospheric change factors combined on this pathosystem. Using controlled environment chambers, plant growth parameters and biochemistry, and aphid life history (development and fecundity) were examined under two different scenarios. Both included climate and atmospheric factors; current conditions (aCO_2_ = 400 ppm, aT = 20 °C) and the predicted (i.e., future climate) conditions by the end of the century, according to the Intergovernmental Panel on Climate Change (IPCC) [1] (eCO_2_ = 800 ppm, eT = 22 °C). To test climate conditions effects on virus infection on wheat and *R. padi*, plants were either non-infected or infected with BYDV-PAV. We hypothesized that combined eCO_2_ and eT could change BYDV-PAV infection on wheat plants differently, impacting plant physiology and biochemistry and consequently, the fitness of the aphid vector.

## 2. Materials and Methods

### 2.1. Biological Material: Plants and Insects

Experiments were conducted at Grains Innovation Park (GIP), Agriculture Victoria, in Horsham, Australia. Wheat plants cv. Yitpi were grown in 7 × 7 × 16 cm (length × width × height) pots filled with potting mix containing slow release fertilizer, trace elements, iron, and lime (12.45 kg/m^3^). Prior to virus inoculation, plants were maintained in growth chambers (Thermoline Scientific, Sydney, Australia, TPG- 1260) at 60–70% relative humidity, 20 °C temperature, 14:10 h (L:D) photoperiod, and light intensity of 1000 μmol m^−2^ s^−1^ at the top of the plant canopy generated by five 400 W high-pressure sodium and four 70 W incandescent globes. Plants were basal watered every day to maintain a standardised watering regime across all treatments.

To initiate the aphid laboratory culture, a single parthenogenetic female of non-viruliferous *R. padi* was obtained from vegetation located at GIP. Laboratory clonal colonies were reared on wheat plants (cv. Yitpi) and maintained in a growth chamber at 20 °C temperature and 14:10 h (L:D) photoperiod. Aphids were synchronized prior to the bioassay to guarantee age homogeneity (two-day-old adults) and were introduced onto four-week-old plants at the time the experiment was initiated.

### 2.2. Virus Isolate and Inoculation

The BYDV-PAV isolate used in the experiment was originally obtained from wild oat (*Avena sativa* L.) collected near GIP facilities. Virus identity was confirmed by tissue blot immunoassay (TBIA), then reverse transcription polymerase chain reaction (RT-PCR) using PAV-specific primers [11]. BYDV-PAV was maintained at the GIP on wheat (cv. Yitpi) in plant growth chambers at a constant 20 °C and 14:10 h (L:D) photoperiod. Every fortnight, virus was transmitted to new plants using *R. padi*, and confirmed using TBIA, similarly as explained below. 

When plants reached the two-leaf stage (11-days old), half the plants sown were inoculated with BYDV-PAV by exposure to 10 viruliferous *R. padi* per plant. The tip of the first leaf (4–4.5 cm long) was inserted into a plastic tube (5 cm height, 1.5 cm diameter) containing the infected aphids and sealed with cotton wool. After 72 h, all aphids were carefully removed [38]. The remainder of plants sown were uninoculated controls [12]. To determine the inoculation success or eliminate cross contamination, all experimental plants were tested to confirm presence or absence of BYDV-PAV at the end of the experiment. Three stems of each plant were cut, blotted onto nitrocellulose membrane (Amersham Potran, 0.45 μm, Germany), and then tested by TBIA [11].

### 2.3. Acclimation of Insects and Plants to Elevated CO_2_ and Temperature

Plants were maintained in two growth chambers (Thermoline Scientific, TPG- 1260) under the same relative humidity (60–70 %), photoperiod (14:10 (L:D)), and light intensity (1000 μmol m^−2^ s^−1^ at the top of the plant canopy) conditions but with different atmospheric CO_2_ concentration and temperature levels. One chamber was set to elevated CO_2_ and temperature (eCO_2_&eT) (800 ppm and 22 °C), while the other was set to ambient CO_2_ and temperature (aCO_2_&aT) (400 ppm and 20 °C). The ambient CO_2_ was measured in Horsham, Victoria, Australia in July 2018, and the ambient temperature is based on the long-term average maximum temperature in Horsham for September and October, when peak abundance of *R. padi* occurs.

After virus inoculation, plants were divided into four sets (*n* = 32/treatment), depending on the CO_2_&T conditions and virus infection treatment: aCO_2_&aT × non-infected plants; aCO_2_&aT _×_ BYDV-PAV-infected plants; eCO_2_&eT × non-infected plants; eCO_2_&eT × BYDV-PAV-infected plants.

To allow the aphids to acclimate to the different environments, non-viruliferous *R. padi* colonies were placed in each chamber with different T and CO_2_ concentration for 2–3 generations prior to the beginning of the experiments [60].

During acclimation and bioassays, CO_2_, temperature, plants, aphid colonies and experimental units were alternated between climate chambers every second day to minimize any potential chamber effects. 

### 2.4. Plant Traits

Plant growth parameters, including plant height, number of tillers, and leaf chlorophyll content, were measured once a week for six consecutive weeks on BYDV-PAV-infected and non-infected plants, grown under current ambient (aCO_2_&aT) or elevated (eCO_2_&eT) conditions. First measurement was done directly after virus inoculation (on the day the viruliferous aphids were removed), and this day was considered zero week post-inoculation (wpi) (*n* = 32/treatment, for the first four weeks; *n* = 20/treatment, for the last two weeks as the rest were used for plant destructive analyses). Chlorophyll content was measured via soil plant analytical development (SPAD) using the SPAD-502Plus chlorophyll meter (Konica Minolta, Japan), commonly used as an indirect measurement of foliar N content which positively correlates leaf chlorophyll and leaf N content [61]. SPAD readings were taken from the fully expanded leaf of the main tiller (stem), which consisted of, on average, three measurements per plant. 

In addition to the weekly plant measurements, the biomass and C:N content of designated aphid-free plants (*n* = 12/treatment) were analysed four weeks after inoculation, when aphids reached adulthood in the corresponding aphid fitness experiment. Leaves and stems were excised then oven dried (TD-150F, Thermoline Scientific, NSW, Australia) at 60 °C for 72 h and subsequently weighed using an analytical scale (PA512C, Ohaus corporation, USA). Dried samples (leaves and stems combined) were finely ground (<0.5 mm) using a ball mill (Retsch MM300, Haan, Germany). Total C and N content of plant tissue samples were determined via the Dumas combustion method using a CHN analyzer (CHN 2000, LECO, St. Joseph, MO, USA) at the University of Melbourne, Creswick, Australia. At the end of the experiment (five weeks post inoculation), wheat growth stage was evaluated by the Zadoks Decimal Code [62] (*n* = 20/treatment).

### 2.5. Rhopalosiphum Padi Fitness Experiment

To assess aphid development and fecundity, three weeks post-inoculation, one single non-viruliferous winged adult female (two days old) from the corresponding acclimated colony (eCO_2_&eT or aCO_2_&aT) was placed on the second fully expanded leaf of the main stem of each wheat plant to generate progeny. The aphid was confined in a clip cage (3 cm diameter with a fine mesh top to allow transpiration) on a transparent acrylic platform (6 × 4 cm) and secured to the plant by a 9-cm-long hair clip. After 24 h, adult females and all nymphs generated except one were removed. The remaining nymph was left on the plant and monitored daily. When the nymph reached adulthood and started to generate offspring, progeny was counted daily, then removed. Aphids were monitored over 11 days while producing offspring. The duration of the four nymphal instars, period from the beginning of adulthood to the onset of reproduction, period from birth to adulthood, pre-reproductive period (period from birth to the onset of reproduction (*d*)), effective fecundity (offspring for a period equal to the pre-reproductive period (*Md*)), total fecundity (offspring number per female over a 11 day period (*M11*)), fecundity per day, intrinsic rate of natural increase (*r_m_ = 0.738 (ln Md)/d*), mean generation time (*Td = d/0.738*), and mean relative growth rate (*RGR = r_m_/0.86*) were calculated. The *r_m_* is a simple estimator of the reproductive potential of an aphid population and is positively related to *RGR*, that is an indirect measure of the quality of the food supply; 0.738 is a correcting constant of the proportion of the total fecundity produced by an aphid female in the first days of reproduction; 0.86 is a conversion factor [63,64]. Each plant with one aphid was considered one repetition (*n* = 20/treatment).

### 2.6. Statistical Analysis

Wheat growth stage, biomass and C and N content, as well as *R. padi* biological parameters, were analyzed using two-way analysis of variance (ANOVA) in order to determine the main effects of the factors, i.e., the CO_2_&T conditions and the virus infection, as well as their interaction. Furthermore, in order to quantify the variations between the different combinations and to contrast how the effect due to virus infection varies between aCO_2_&aT to eCO_2_&eT and the effect due to CO_2_&T conditions changes when compared non-infected to BYDV-PAV-infected plants, pairwise multiple comparisons were run thereafter to identify the simple main effects at a confidence interval adjustment of the least significant difference (LSD). A significance level of α = 5% was considered (*p* < 0.05). When needed, parameters were transformed (log(x + 1)) to fit normality and homoscedasticity. These statistical tests were conducted using the general linear model (GLM) module in IBM SPSS Statistics 22.0.0.0 software for Windows (Chicago, IL, USA). 

To analyze the effect of the factors on weekly plant growth parameters and daily fecundity of *R. padi*, we used a linear mixed-effect model (LMM) with CO_2_&T conditions and virus infection as fixed factors, and period (week or day, depending on the case) as the repeated measures factor. The best covariance structure for the repeated measures factor was that with the lowest value of Akaike information criterion (AIC) [65]. Whenever interaction between factors was significant (*p* < 0.05), LSD confidence interval adjustment was performed to compare the estimated marginal means.

## 3. Results

### 3.1. Plant Growth Parameters

There was a statistically significant interaction between CO_2_&T conditions and virus infection on plant height depending on the week (CO_2_&T × virus × week: F_5,98_ = 2.558, *p* = 0.032). The heights of BYDV-PAV-infected plants was significantly lower than non-infected plants in both CO_2_&T conditions from 1 wpi under aCO_2_&aT and 2 wpi under eCO_2_&eT onward (*p* ≤ 0.05 in all cases). There were no significant differences in height comparing virus infected plants under eCO_2_&eT to those under aCO_2_&aT, but the heights of non-infected plants at 4 wpi and 5 wpi were significantly different comparing eCO_2_&eT to aCO_2_&aT (4 wpi: F_1,108_ = 8.197, *p* = 0.005; 5 wpi: F_1,107_ = 14.059, *p* < 0.001) (Figure 1a). 

There was a statistically significant interaction among CO_2_&T conditions and virus infection on the number of tillers depending on the week (CO_2_&T × virus × week: F_5,158_ = 2.616, *p* = 0.027). The number of tillers was significantly higher under eCO_2_&eT compared to aCO_2_&aT for both non-infected and virus-infected plants since 1 wpi (*p* ≤ 0.05 in all cases, except when compared eCO_2_&eT to aCO_2_&aT in BYDV-PAV-infected plants at 2 wpi: F_1,124_ = 1.423, *p* = 0.235). Significantly more tillers were observed in non-infected than BYDV-PAV-infected plants under eCO_2_&eT from 2 wpi until 4 wpi (2 wpi: F_1,124_ = 7.312, *p* = 0.008; 3 wpi: F_1,124_ = 4.791, *p* = 0.030; 4 wpi: F_1,78_ = 7.562, *p* = 0.007) (Figure 1b). 

There was a statistically significant interaction among CO_2_&T conditions and virus infection on chlorophyll content depending on the week (CO_2_&T × virus × week: F_5,129_ = 7.107, *p* < 0.001). A decreasing trend in SPAD, which shows changes in chlorophyll content (higher values represent greener plants), was observed since 2 wpi. Chlorophyll content was significantly lower in BYDV-PAV-infected than non-infected plants, irrespective of CO_2_&T conditions, since 2 wpi (*p* ≤ 0.05 in all cases) (Figure 1c). Moreover, yellow leaf discoloration, which represents typical symptoms of BYDV infection, was more visible in BYDV-PAV-infected plants under eCO_2_&eT than under aCO_2_&aT since 1 wpi (*p* ≤ 0.05 in all cases) (Figure 1c). 

Plant growth stage (5 wpi) was significantly affected by CO_2_&T conditions depending on the virus infection (CO_2_&T × virus: F_1,76_ = 11.609, *p* = 0.001). Non-infected plants matured 35% faster and BYDV-PAV-infected plants matured 20% faster under eCO_2_&eT than under aCO_2_&aT. Under both CO_2_&T conditions, virus infection significantly delayed plant growth, although the difference between infected and non-infected plants was higher under eCO_2_&eT (24%) than those under aCO_2_&aT (14%) (Figure 2). 

Overall, biomass (dry weight) of six-week-old plants (4 wpi) significantly increased under eCO_2_&eT (F_1,44_ = 111.925, *p* < 0.001) and decreased due to virus infection (F_1,44_ = 13.595, *p* = 0.001) (Figure 3). Specifically, plant biomass under eCO_2_&eT compared to under aCO_2_&aT significantly increased by 63% on non-infected wheat (F_1,44_= 52.773, *p* < 0.001) and by 88% on BYDV-PAV-infected wheat (F_1,44_= 59.245, *p* < 0.001). Virus infection significantly reduced 24% plant biomass under aCO_2_&aT (F_1,44_= 7.972, *p* = 0.007) and 13% under eCO_2_&eT (F_1,44_= 5.716, *p* = 0.021) (Figure 3). 

### 3.2. Plant Carbon and Nitrogen Content

The CO_2_&T conditions had a significant effect on the content of C, N, and C:N ratio of wheat plants, whereas virus infection had a statistically significant effect only on C:N ratio (Figure 4). 

Carbon content significantly increased in wheat plants under eCO_2_&eT than those under aCO_2_&aT (F_1,42_ = 131.265, *p* < 0.001) (Figure 4a), specifically, by 5% in non-infected plants (F_1,42_ = 55.037, *p* < 0.001) and by 6% in BYDV-PAV-infected plants (F_1,42_ = 77.764, *p* < 0.001) (Figure 4a).

Nitrogen content significantly decreased in wheat plants under eCO_2_&eT than those under aCO_2_&aT (F_1,42_ = 108.098, *p* < 0.001), in particular, by 35% in non-infected plants (F_1,42_ = 64.580, *p* < 0.001) and by 28% in BYDV-PAV-infected plants (F_1,42_ = 44.098, *p* < 0.001). Overall, no differences in N content due to virus infection were observed (F_1,42_ = 3.590, *p* = 0.065) (Figure 4b); however, under eCO_2_&eT, BYDV-PAV-infected plants significantly increased N content by 13% compared to non-infected plants (F_1,42_ = 4.827, *p* = 0.034). 

The C:N ratio significantly increased in wheat plants under eCO_2_&eT than those under aCO_2_&aT (F_1,42_ = 110.131, *p* < 0.001) and significantly decreased in BYDV-PAV-infected plants compared to non-infected plants (F_1,42_ = 4.715, *p* = 0.036) (Figure 4c). Due to eCO_2_&eT exposure, C:N ratio increased significantly in non-infected plants by 60% (F_1,42_ = 66.583, *p* < 0.001) and in virus infected plants by 49% (F_1,42_ = 44.251, *p* < 0.001) respectively, when compared to aCO_2_&aT. Furthermore, C:N ratio was significantly reduced by 11% due to virus infection under eCO_2_&eT (F_1,42_ = 6.003, *p* = 0.019).

### 3.3. Rhopalosiphum Padi Fitness Experiment

Overall, aphid fecundity and development time parameters were significantly affected by CO_2_&T conditions and by the virus infection (Table 1 and Figure 5 and Figure 6).

Aphid development time from birth to adulthood (period from nymphal birth to adult exuvia removal) was significantly shorter under eCO_2_&eT compared to aCO_2_&aT (F_1,75_ = 20.260, *p* < 0.001), both on non-infected plants (F_1,75_ = 17.322, *p* < 0.001) and on BYDV-PAV-infected plants (F_1,75_ = 4.912, *p* = 0.030); and significantly increased when aphids were reared on BYDV-PAV-infected plants compared than those on non-infected plants (F_1,75_ = 12.436, *p* = 0.001) (Table 1). Thus, this development time increased due to virus infection under each CO_2_&T condition but it was significantly longer only under eCO_2_&eT (F_1,75_ = 12.029, *p* = 0.001).

*Rhopalosiphum padi* pre-reproductive period (*d*) and mean generation time (*Td* = *d*/0.738) significantly decreased under eCO_2_&eT when compared to aCO_2_&aT (*d*: F_1,75_ = 4.046, *p* = 0.048; *Td*: F_1,75_ = 4.049, *p* = 0.048), and were significantly longer due to virus infection (*d*: F_1,75_ = 8.007, *p* = 0.006; *Td*: F_1,75_ = 8.010, *p* = 0.006) (Table 1). When compared to aCO_2_&aT, eCO_2_&eT decreased *R. padi d* and *Td*, although the parameters were not statistically different on non-infected plants (*d*: F_1,75_ = 3.423, *p* = 0.068; *Td*: F_3,75_ = 3.433, *p* = 0.068) nor virus infected plants (*d*: F_1,75_ = 1.000, *p* = 0.321; *Td*: F_3,75_ = 0.997, *p* = 0.321). Within each CO_2_&T condition, *d* and *Td* increased due to virus infection but were significantly longer only under eCO_2_&eT (*d*: F_1,75_ = 5.917, *p* = 0.017; *Td*: F_1,75_ = 5.929, *p* = 0.017).

Generally, the development time of each nymphal instar remained unchanged due to CO_2_&T conditions as well as virus infection, except for the second nymphal stage (N2). The duration of N2 significantly decreased under eCO_2_&eT when compared to aCO_2_&aT (F_1,75_ = 14.196, *p* < 0.001) and was significantly longer due to virus infection (F_1,75_ = 10.592, *p* = 0.002). The period from the beginning of adulthood to the onset of reproduction was affected by CO_2_&T conditions and increased under eCO_2_&eT when compared to aCO_2_&aT (F_1,75_ = 4.742, *p* = 0.033) (Appendix A). 

Fecundity of *R. padi* per day fluctuated throughout the experiment with a similar trend under each of the different CO_2_&T and virus conditions. There was a statistically significant interaction among the effects of CO_2_&T conditions and virus infection on the fecundity depending on the day and, in general, fewer number of nymphs was produced by *R. padi* under eCO_2_&eT than under aCO_2_&aT (LMM, CO_2_&T × virus × day: F_10,73_ = 2.133, *p* = 0.032) (Figure 5). There was a significant decrease in *R. padi* effective fecundity (*Md*) under eCO_2_&eT, when compared with those reared under aCO_2_&aT conditions (F_1,73_ = 31.292, *p* < 0.001), specifically by 13% on non-infected plants (F_1,73_ = 9.492, *p* = 0.003) and by 18% on virus infected plants (F_1,73_ = 23.020, *p* < 0.001). Overall *Md* was significantly higher due to virus infection (F_1,73_ = 16.478, *p* < 0.001) (Table 1); however, under each CO_2_&T condition, fecundity was significantly higher (by 17%) under aCO_2_&aT (F_1,73_ = 14.286, *p* < 0.001) but not under eCO_2_&eT (F_1,73_ = 3.796, *p* = 0.055). When comparing fecundity per female over the whole experimental period (*total fecundity*, *M11*), the same trend was observed as described for *Md* and statistically significant differences were observed due to CO_2_&T conditions and virus infection (CO_2_&T: F_1,73_ = 29.974, *p* < 0.001; virus: F_1,73_ = 11.688, *p* = 0.001) (Figure 6).

The intrinsic rate of natural increase (*r_m_*) and the mean relative growth rate (*RGR*), which are parameters related by the formula [*r_m_* = 0.86 *RGR*], did not differ significantly due to CO_2_&T conditions nor virus infection (*r_m_* and *RGR*: CO_2_&T: F_1,75_ = 0.179, *p* = 0.674; virus: F_1,75_ = 1.715, *p* = 0.194; CO_2_&T × virus: F_1,75_ = 0.164, *p* = 0.687) (Table 1). 

Significant effects of climate and atmospheric conditions and BYDV-PAV infection on wheat traits and aphid biological parameters are summarized in Figure 7.

## 4. Discussion

By the end of this century, atmospheric CO_2_ concentration and global temperature on Earth will increase [1], directly and indirectly altering plant, insect, and virus interactions, thus affecting ecosystems, agriculture, and food production [44]. To date, most of the research regarding the impact of climate change on plant–insect–pathogen interactions has been conducted on a single climate or atmospheric variable, mainly either temperature or CO_2_ [13,14,47,48,66]. However, to create representative conditions of future climate, research approaches should include the responses of multitrophic systems to climate variables that will occur simultaneously [17,27,45]. To our knowledge, this is the first empirical study describing the interactions between wheat and its economically important aphid pest and viral pathogen under combined eCO_2_ and eT.

We have found that eCO_2_&eT directly affects wheat growth depending on virus infection, and that eCO_2_&eT and virus infection directly influences the rest of wheat plants parameters, similar to when evaluating the effect of eCO_2_ solely for the same wheat cultivar [14]. However, when evaluating the vector fitness of the pathosystem, the *R. padi* growth rate under eCO_2_&eT is similar to when grown under ambient conditions, so the combination of the two climatic factors could have a neutral effect on *R. padi* fitness when reared on non-infected or BYDV-PAV-infected plants, contrarily to the reduction in the growth rate for *R. padi* developed on non-infected wheat under the sole effect of eCO_2_ [14].

All plant parameters analyzed in our study (except number of tillers) were influenced by BYDV-PAV infection, but the effect of eCO_2_&eT differed depending on the plant variable. Typical virus symptoms, such as leaf discoloration (chlorophyll content reduction), developed earlier and were more distinct under eCO_2_&eT than under aCO_2_&aT conditions, which is consistent with other findings under higher temperature [43] or eCO_2_ [12,15]. Our results showed a significant increase in the number of tillers on wheat plants grown under eCO_2_&eT compared to those under aCO_2_&aT but a decrease in the number of tillers due to virus infection under eCO_2_&eT, as has been similarly described when BYDV-wheat plants were grown under eCO_2_ [12,67] or under eT [43]. Hence, future climate can exacerbate the BYDV symptomatology in wheat.

The fertilising effect of eCO_2_ that increases biomass and yield is commonly recognised in wheat [68], oat [15], and other crops [13,18,22,69]. Increased plant biomass and BYDV-PAV titre have been shown to occur not only under eCO_2_ [38] but also under eT [43] in wheat grown under controlled conditions. However, a study performed in open-top chambers indicated that wheat biomass and yield increases depend on eCO_2_ but not as a result of eT nor of the interaction [34]. In our study, biomass increased in non-infected and BYDV-PAV-infected wheat under eCO_2_&eT, and there was a significant reduction in biomass due to virus infection under eCO_2_&eT and under aCO_2_&aT. In other studies, the positive effect of CO_2_ fertilization has been offset by increasing temperatures, therefore wheat grain yield and biomass could be reduced in the future [70,71]. Thus, the potential increase in biomass stimulated by eCO_2_ is only achieved when plants develop within the interval of optimal temperature for their growth [72]. The frequency and amplitude of extreme high temperatures and heat waves may impact differently in plant and higher trophic levels [73,74]. 

Elevated CO_2_ increases plant growth rate [68,69]. In our study, eCO_2_ combined with higher temperature also increased the growth rate of non-infected and BYDV-PAV-infected wheat. Nevertheless, BYDV infection prolonged wheat development time and virus symptomatology were more distinct under eCO_2_&eT. This could have additional implications on disease epidemiology, wheat yield and quality in the future. 

Reduction in N content has been observed in multiple plant species under eCO_2_ [12,13,14,22,23,25,27,69] and under eCO_2_&eT combined [40], as-in our study. Nitrogen is essential for aphid biological functions [24,75] therefore, the reduction in *R. padi* fecundity under eCO_2_&eT could be attributed to the changes in the quantity and quality of plant N [41,44]. In our study, overall fecundity significantly decreased and development time to reach adulthood and pre-reproductive period were shorter under eCO_2_&eT than aCO_2_&aT; however, *R. padi* growth rate was not significantly different under eCO_2_&eT when compared to aCO_2_&aT. Decreased fecundity can indicate lower aphid population levels, but if it is offset by a shorter development time, as shown here, we are inclined to conclude that aphid numbers will likely remain unchanged in the future. Growth rates (*r_m_* or RGR) refer to the rate at which a population grows and balances the relationship between fecundity and development time; therefore, no differences in our *R. padi* growth rates indicate that there will be no change in the speed of growth of the aphid population under the future climate and atmospheric conditions that we established. *Rhopalosiphum padi* is currently a major aphid pest in temperate cereal crops worldwide [59]. If the population level remains unchanged, this aphid will probably continue to have a significant economic importance in the future. Hoover and Newman [31] reported that the predicted negative effects of eCO_2_ on herbivores could be offset by eT, and that the responses of cereal aphid populations to the two interrelated factors will be more similar to current ambient conditions than the expected future conditions when examining the factors separately. It further showed that the divergence in cereal aphid performance could be based on N limitation. Large N availability under eCO_2_, due to high soil N fertility and low aphid N requirements, could probably increase aphid populations; however, eT would reduce this effect [31,37]. Other studies analyzing the effect of eCO_2_ singly showed a decline in abundance of *R. padi* under eCO_2_ in a pasture grass [32]. Moreover, Trȩbicki et al. (2016) showed a reduction in *R. padi* fecundity and in growth rates reared on non-infected wheat under eCO_2_, which could indicate lower aphid pest pressure. Conversely, other research showed a different trend in *R. padi* population on wheat, with an increase in fecundity therefore population under eCO_2_ [76] or an increase in the population under eCO_2_ and eT combined [34]. Furthermore, under eT and eCO_2_, *R. maidis* (Fitch) increased its performance and produced higher number of winged aphids, which could increase the spread of plant viruses and exacerbate aphid damage on barley crops [35]. Overall, these investigations underline that making generalizations of the effects of climate change factors on multitrophic interactions remains difficult [2,44]. Furthermore, different growing conditions, cultivars and species/genotypes, eCO_2_ concentration and temperature increases, etc. could explain the contrasting results and highlight potential aphid/host plant specificity. 

A virus can modify host plants and vector behavior to enhance its spread under ambient conditions [77,78,79,80]. Cereal aphids reared on BYDV-infected plants have significantly shorter development time and/or increased fecundity compared to non-infected plants [81,82,83]. Winged cereal aphids are visually attracted to BYDV-infected plants [83,84]. Furthermore, *R. padi* increases alate production particularly when reared on virus-infected plants [46]. Therefore, vector performance can improve on virus-infected plants and its preference for virus-infected plants could promote virus spread [82]. Elevated CO_2_ affects symptomatology of plant viruses, enhancing earlier or greater differences (leaf discoloration) in infected plants, making them more attractive to insect vectors [13,38]. Elevated CO_2_ can alter the epidemiology of BYDV by increasing the virus persistence of infected oat plants [15] and the virus incidence in wheat grown under Free Air Carbon Dioxide Enrichment Facility (FACE) [11]. Furthermore, eCO_2_ also affects *R. padi* feeding behavior and therefore the potential for virus acquisition and transmission efficiency [14]. The effects of eCO_2_ on virus infection can be pathosystem- or even genotype-specific [13,85]. In addition, increased temperature can impact virus incidence, resulting in an earlier and greater BYDV-PAV titer in wheat [43]. The efficiency of *R. padi* to transmit BYDV-PAV usually increases as temperature rises inside the optimal thermal thresholds [46,86]. In our research, BYDV-PAV infection altered wheat growth and C:N ratio under the combination of eCO_2_ and eT. Under eCO_2_, virus infection also increased N content in BYDV-infected plants, which can explain greater host suitability for *R. padi* [14]. However, no effect on higher trophic level was shown in our study, as *R. padi* growth rate remained unchanged when aphids were grown on BYDV-PAV-infected plants under eCO_2_&eT compared to aCO_2_&aT. The earlier virus symptomatology found under eCO_2_&eT could make virus-infected plants more attractive to aphids. Within the same pathosystem that we analyzed, BYDV-PAV titer in wheat plants increased under eCO_2_ [38] and under eT [43] when assessed separately, which could intensify the spread of BYDV disease by increasing virus transmission and acquisition efficiency.

We hypothesize that the neutral effect on *R. padi* growth rate as shown in our study is attributed to the combined effects of temperature and CO_2_, as an increase in temperature usually shortens insect development time while eCO_2_ indirectly affects aphid fecundity due to the lower nutrition quality of the host plant. Further research is needed to elucidate the impact of the combination of eT and eCO_2_ on the pathosystem considered by studying the attractiveness of *R. padi* to non-infected and virus-infected plants and by examining the virus titer in order to understand the potential virus dispersion and transmission under future climate and atmospheric conditions. Moreover, other abiotic factors, such as water stress or ozone, should be considered, to allow a more realistic prediction of the impact on food security due to the variations on pests and diseases under changing climate. 

## 5. Conclusions

Plant, insect herbivore, and pathogen responses to climate change can be specific to each level of the multitrophic system [19] and to the interactions among them [87]. Under a climate change scenario, the effects on the pathosystem could be modified or even mitigated because of the interrelation of the climate and atmospheric change factors, and the impact of the combination could be different to the impact of the abiotic factors analyzed separately [2,40]. Our research supports the increasing concern that investigating specific climatic factors in isolation might not be representative of the complex interactions on aphid–plant–pathogen interactions in agro-systems under the future climate [17].

Our study shows that simultaneous increase in CO_2_ and temperature as predicted by 2100 will impact a particular plant–vector–virus system differently depending on virus infection. Under eCO_2_&eT, virus symptomatology in wheat developed earlier, which could increase visual attraction of *R. padi* to BYDV-infected plants and lead to an increase in virus spread prompting additional implications on disease epidemiology, therefore affecting wheat yield and quality in the future. 

Nitrogen is essential for aphid biological functions thus, the decrease in *R. padi* fecundity under eCO_2_&eT could be attributed to the reduction in plant N content. BYDV-PAV infection compensated partially this N dilution and increased *R. padi* fecundity and development time. However, independently of virus infection, aphid growth rates did not differ due to climate conditions. As *R. padi* is currently a major aphid pest in temperate cereal crops worldwide, if its population levels remain unchanged in the future, this aphid species will continue to have a significant economic importance, mainly due to its role as a virus vector. 

This type of forecast studies highlights the importance of the effects of climate change on virus vectors and provides important information which can aid in the development of effective crop protection strategies to ensure the global food security. 

## Figures and Tables

**Figure 1 insects-11-00459-f001:**
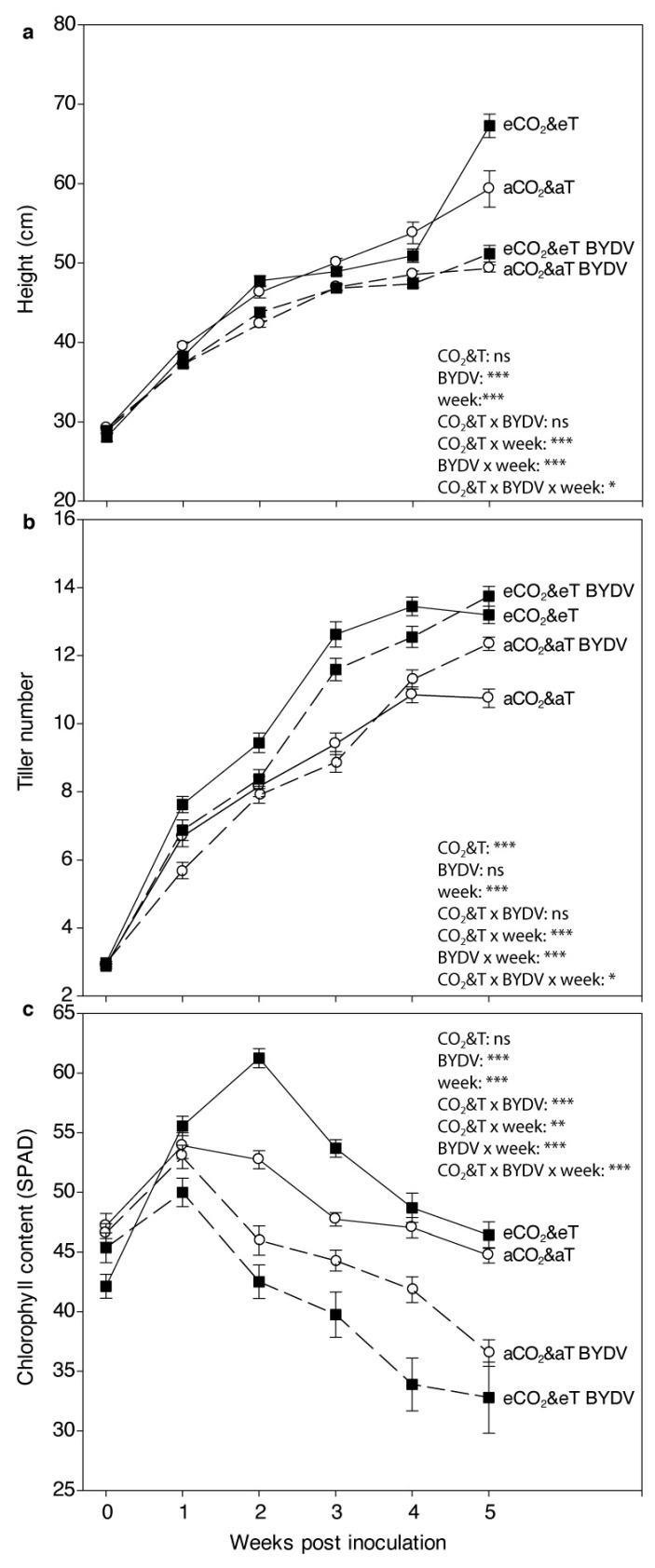
Weekly plant growth parameters. (**a**) Height (cm), (**b**) number of tillers, and (**c**) chlorophyll content (SPAD) of non-infected (aCO_2_&aT or eCO_2_&eT) and BYDV-PAV-infected (aCO_2_&aT BYDV or eCO_2_&eT BYDV) wheat plants developed under aCO_2_&aT (400 ppm and 20 °C) or eCO_2_&eT (800 ppm and 22 °C). Linear mixed-effect model with CO_2_&T conditions and virus infection (BYDV) as fixed factors, and week as the repeated measures factor. Significant differences represented by asterisks: * (*p* ≤ 0.05), ** (*p* ≤ 0.01), *** (*p* ≤ 0.001), ns: not significant. Error bars represent standard error (SE). *n* = 32/treatment, for 0–3 weeks post inoculation; *n* = 20/treatment, for 4–5 weeks post inoculation.

**Figure 2 insects-11-00459-f002:**
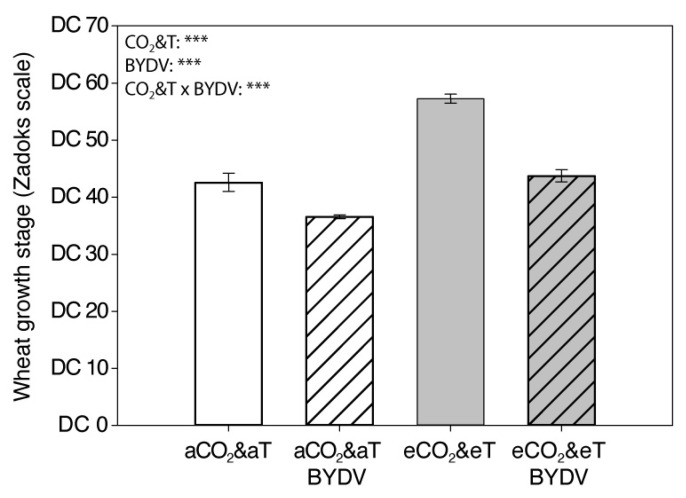
Wheat growth stage five weeks post inoculation. Growth stage of non-infected (aCO_2_&aT or eCO_2_&eT) and BYDV-PAV-infected (aCO_2_&aT BYDV or eCO_2_&eT BYDV) 7-week-old plants (5 weeks post inoculation) grown under aCO_2_&aT (400 ppm and 20 °C) or eCO_2_&eT (800 ppm and 22 °C). Wheat growth stage according to Zadok decimal code (DC30-36: stem elongation; DC37-49: flag leaf to booting; DC51-60: heading; DC61-69: flowering) [62]. *p*-values according to two-way ANOVA test. Significant differences represented by asterisks: *** (*p* ≤ 0.001). Error bars represent standard error (SE). *n* = 20.

**Figure 3 insects-11-00459-f003:**
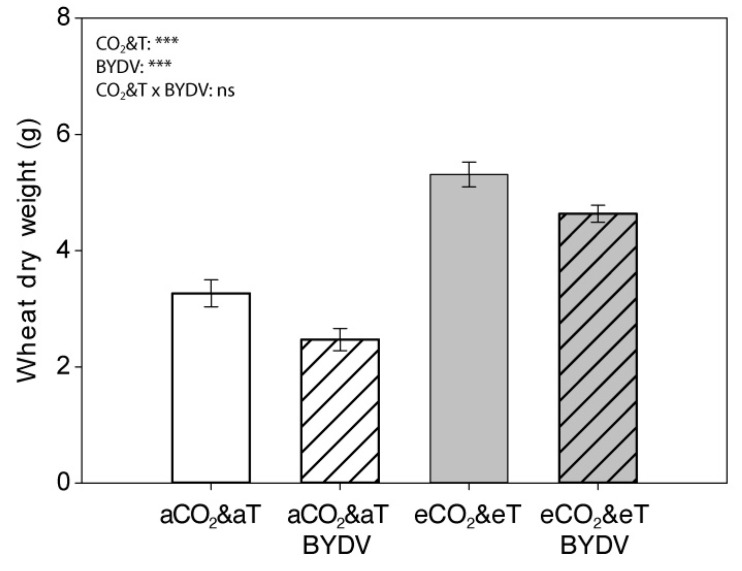
Wheat above-ground biomass four weeks after inoculation. Dry weight (g) of 6-week-old (4 weeks post inoculation) non-infected (aCO_2_&aT or eCO_2_&eT) and BYDV-PAV-infected (aCO_2_&aT BYDV or eCO_2_&eT BYDV) wheat plants (leaves and stems pooled) grown under aCO_2_&aT (400 ppm and 20 °C) or eCO_2_&eT (800 ppm and 22 °C). *p*-values according to two-way ANOVA test. Significant differences represented by asterisks: *** (*p* ≤ 0.001), ns: not significant. Error bars represent standard error (SE). *n* = 12.

**Figure 4 insects-11-00459-f004:**
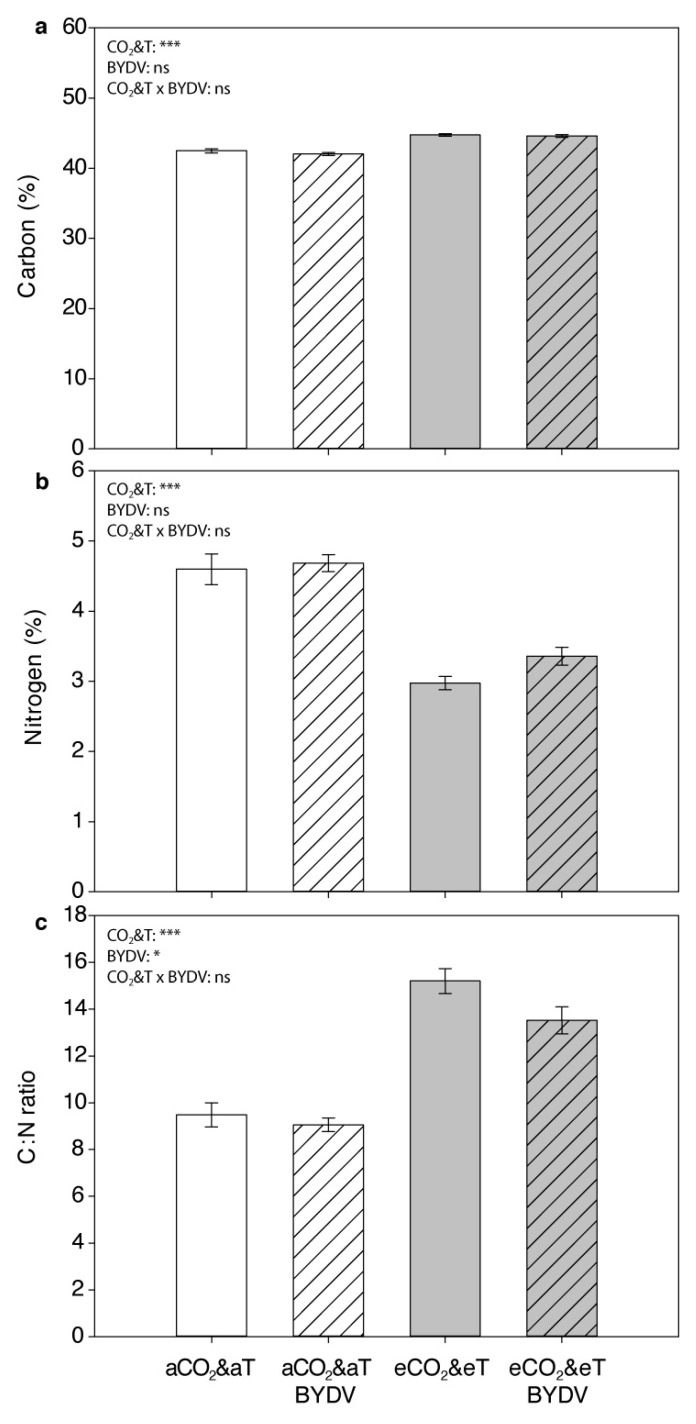
Wheat chemical profile. (**a**) Carbon (C) concentration (%), (**b**) Nitrogen (N) concentration (%) and (**c**) C:N ratio of above-ground plant parts (leaves and stems pooled) of 6-week-old (4 weeks post inoculation) non-infected (aCO_2_&aT or eCO_2_&eT) and BYDV-PAV-infected (aCO_2_&aT BYDV or eCO_2_&eT BYDV) wheat plants grown under aCO_2_&aT (400 ppm and 20 °C) or eCO_2_&eT (800 ppm and 22 °C). *p*-values according to two-way ANOVA test (N concentration transformed by log (x + 1)). Significant differences represented by asterisks: * (*p* ≤ 0.05), *** (*p* ≤ 0.001), ns: not significant. Error bars represent standard error (SE). *n* = 12.

**Figure 5 insects-11-00459-f005:**
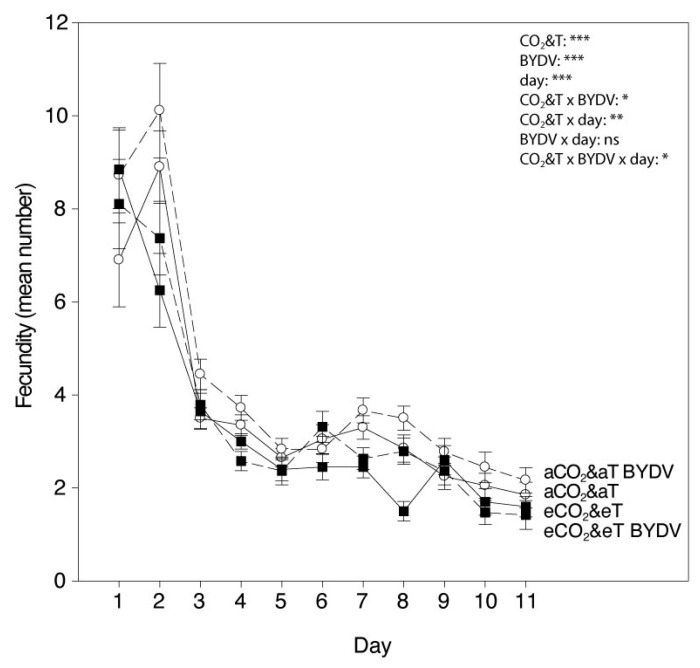
*Rhopalosiphum padi* fecundity per day. Fecundity per day of *R. padi* fed on non-infected (aCO_2_&aT or eCO_2_&eT) and BYDV-PAV-infected (aCO_2_&aT BYDV or eCO_2_&eT BYDV) wheat plants grown under aCO_2_&aT (400 ppm and 20 °C) or eCO_2_&eT (800 ppm and 22 °C). Measured by the daily count of newly emerged nymphs, where day 1 indicates the day when aphids started generating offspring. Linear mixed-effect model with CO_2_&T conditions and virus infection (BYDV) as fixed factors, and day as the repeated measures factor. Significant differences represented by asterisks: * (*p* ≤ 0.05), ** (*p* ≤ 0.01), *** (*p* ≤ 0.001), ns: not significant. Error bars represent standard error (SE). *n* = 20.

**Figure 6 insects-11-00459-f006:**
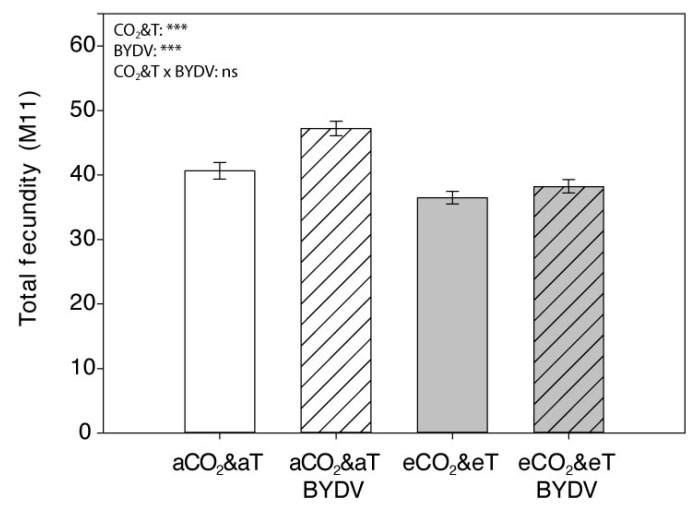
*Rhopalosiphum padi* total fecundity. Mean total fecundity per *R. padi* over an 11-day period, reared on non-infected (aCO_2_&aT or eCO_2_&eT) and BYDV-PAV-infected (aCO_2_&aT BYDV or eCO_2_&eT BYDV) wheat plants grown under aCO_2_&aT (400 ppm and 20 °C) or eCO_2_&eT (800 ppm and 22 °C). *p*-values according to Two-way ANOVA test (log (x + 1) transformation). Significant differences represented by asterisks: *** (*p* ≤ 0.001), ns: not significant. Error bars represent standard error (SE). *n* = 20.

**Figure 7 insects-11-00459-f007:**
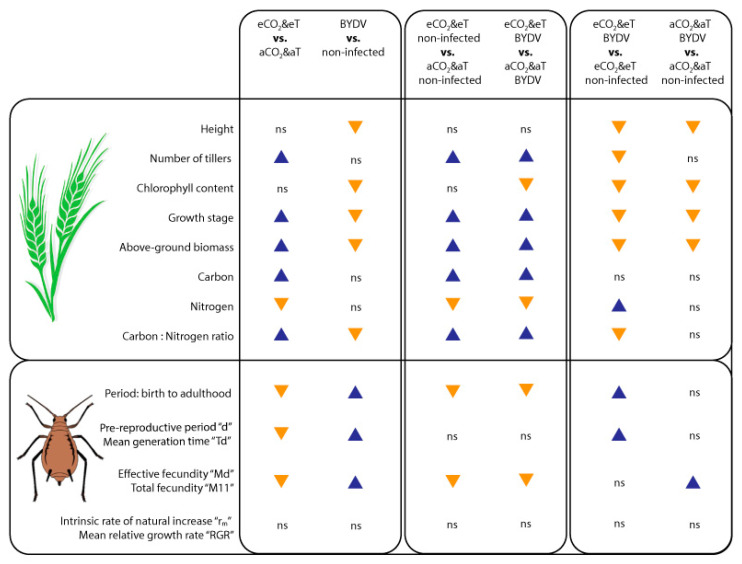
Summary of the effects of CO_2_&T conditions and BYDV-PAV infection on wheat traits and on *R. padi* biological parameters. *Rhopalosiphum padi* was reared on non-infected and BYDV-PAV-infected wheat plants grown under aCO_2_&aT (400 ppm and 20 °C) or eCO_2_&eT (800 ppm and 22 °C) conditions. Weekly plant growth parameters (height, number of tillers and chlorophyll content) were analysed by the linear mixed-effect model; the rest of parameters by two-way ANOVA; pairwise multiple comparisons were conducted thereafter to identify the simple main effects (variation within each factor that depends on the level of the other factor) at an LSD confidence interval adjustment (*p* ≤ 0.05). Statistically significant increase (blue triangle pointing up), decrease (yellow triangle pointing down) and non-significant differences (“ns”) are indicated.

**Table 1 insects-11-00459-t001:** Life history parameters of *Rhopalosiphum padi* (mean values) reared on non-infected or BYDV-PAV-infected plants under ambient CO_2_&T (aCO_2_&aT = 400 ppm; 20 °C) or elevated CO_2_&T (eCO_2_&eT = 800 ppm; 22 °C).

Aphid Parameter	CO_2_&T	Virus Infection		Mean	SEM	*p*-Value		
		Non-Infected	BYDV-PAV Infected			CO_2_&T	Virus	CO_2_&T × Virus
*Period from birth to adulthood (days)*	aCO_2_&aT	7.250	7.474	7.362 ^A^	0.103	**<0.001 *****	**0.001 *****	0.182
	eCO_2_&eT	6.650	7.150	6.900 ^B^				
	Mean	6.950 ^b^	7.312 ^a^					
*Pre-reproductive period “d” (days)*	aCO_2_&aT	8.300	8.579	8.439 ^A^	0.121	**0.048 ***	**0.006 ****	0.558
	eCO_2_&eT	8.000	8.400	8.200 ^B^				
	Mean	8.150 ^b^	8.489 ^a^					
*Effective fecundity “Md”*	aCO_2_&aT	35.250	41.111	38.181 ^A^	1.089	**<0.001 *****	**<0.001 *****	0.190
	eCO_2_&eT	30.600	33.579	32.089 ^B^				
	Mean	32.925 ^b^	37.345 ^a^					
*Mean generation time “Td” (days)*	aCO_2_&aT	11.247	11.625	11.436 ^A^	0.164	**0.048 ***	**0.006 ****	0.556
	eCO_2_&eT	10.840	11.382	11.111 ^B^				
	Mean	11.043 ^b^	11.503 ^a^					
*Intrinsic rate of natural increase “r_m_”*	aCO_2_&aT	0.316	0.305	0.311	0.013	0.674	0.194	0.687
	eCO_2_&eT	0.316	0.295	0.305				
	Mean	0.316	0.300					
*Mean relative growth rate “RGR”*	aCO_2_&aT	0.368	0.355	0.361	0.015	0.674	0.194	0.687
	eCO_2_&eT	0.368	0.343	0.355				
	Mean	0.368	0.349					

*p*-values according to two-way ANOVA test for normal and homoscedastic variables. Log (x + 1) transformation to fit homogeneity of variances in some variables. Significant differences represented by asterisks: * (*p* ≤ 0.05), ** (*p* ≤ 0.01) and *** (*p* ≤ 0.001). Different lower-case letters within the row indicate differences due to virus infection (*p* ≤ 0.05). Different upper-case letters within the column indicate differences due to CO_2_&T conditions (*p* ≤ 0.05). SEM: standard error of means. *n* = 20.

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
