# Peer review of "Simultaneous Increase in CO2 and Temperature Alters Wheat Growth and Aphid Performance Differently Depending on Virus Infection"

_insects, 2020, doi:10.3390/insects11080459_

Round 1

Reviewer 1 Report

The manuscript insects-862061 evaluates the effect of different abiotic parameters such as different temperatures (aT vs eT) and COconditions (aCOvs eCO2) in non-infected wheat and BYDV-infected wheat and also in BYDV vector biological fitness. The results of this manuscript are sound and original. And the methodology used is correct. In my opinion, this manuscript provides new data to understand how climate change can affect crops and also plant-virus-vector interactions. This is a new area of research that has been paid little attention and this data provides very interesting lines of research for the future. I have only minor edits/typos to be considered:

  • Line 127. Place the same spaces between 7x7x16 cm.
  • Line 154. Change CO2 to CO2.
  • Lines 298 and 302. Change “Figure4a” to “Figure 4a”.
  • Table 1. Prereproductive period. Is the abbreviature “d” repeated?. Please, verify that the abbreviatures used in material and methods (lines 204-209) are the same as the ones in Table 1. I am confused why the use of (d) (d) for pre-reproductive period instead of only (d) and why the use of (Td) (d) instead of only (Td) in mean generation time.
  • Figure 6: in the Y axes write fecundity without the space: fecundity instead of "f ecundity".
  • Lines 383: can you explain the biological meaning of the intrinsic rate of natural increase in this paragraph?. It is difficult to understand the meaning of these results. Please, rephrase this paragraph.

Author Response

Reviewer 1

Comments and Suggestions for Authors

The manuscript insects-862061 evaluates the effect of different abiotic parameters such as different temperatures (aT vs eT) and CO2 conditions (aCO2 vs eCO2) in non-infected wheat and BYDV-infected wheat and also in BYDV vector biological fitness. The results of this manuscript are sound and original. And the methodology used is correct. In my opinion, this manuscript provides new data to understand how climate change can affect crops and also plant-virus-vector interactions. This is a new area of research that has been paid little attention and this data provides very interesting lines of research for the future. I have only minor edits/typos to be considered:

Line 127. Place the same spaces between 7x7x16 cm.

Done.

Line 154. Change CO2 to CO2.

Done.

Lines 298 and 302. Change “Figure4a” to “Figure 4a”.

Done. Also corrected in L303.

Table 1. Prereproductive period. Is the abbreviature “d” repeated?. Please, verify that the abbreviatures used in material and methods (lines 204-209) are the same as the ones in Table 1. I am confused why the use of (d) (d) for pre-reproductive period instead of only (d) and why the use of (Td) (d) instead of only (Td) in mean generation time.

We have clarified in Table 1 that the second (d) is for the unit “days” and we have substituted the brackets to quotation marks for the abbreviations: d, Td, Md, rm, RGR.

Furthermore, we have also changed “d” to “days” in the Table S1, and incorporated minor changes in the title and caption of this table.

Figure 6: in the Y axes write fecundity without the space: fecundity instead of "f ecundity".

We have checked our original figure and there is no space.

Lines 383: can you explain the biological meaning of the intrinsic rate of natural increase in this paragraph? It is difficult to understand the meaning of these results. Please, rephrase this paragraph.

Regarding this comment and another related comment of Reviewer 2, we have improved the explanation of the aphid biological parameters in Materials & Methods section (L205-214)

Reviewer 2 Report

The manuscript “Simultaneous increase in CO2 and temperature alters wheat growth and aphid performance differently 3 depending on virus infection” investigates the combined effect of CO2 and temperature on a plant-insect-virus relationship represented by wheat, Barley yellow dwarf virus (BYDV), and the aphid Rhopalosiphum padi. Wheat traits and aphid behaviour were analysed in presence and absence of virus infection and under two CO2 and T conditions, that represent current and future climatic scenarios. The same trophic interaction was already studied under different CO2 and T conditions separately, and the manuscript provides the first results obtained combining the two abiotic factors. For this reason, this study can provide new information on the possible evolution of the viral disease incidence in a future scenario and is worthy to be considered for publication. However, the manuscript is presented as a narrative (like a thesis) and would be better if presented in a scientific way, clearly pointing out the main outcomes that are relevant for final conclusions. The authors should avoid over information, and turn attention to the impact of their findings on the BYDV disease epidemiology in the future. Due to the great amount of data presented in the Results, I also suggest introducing a concluding figure that visually resumes the significant and non-significant effects of CO2 and T conditions. This would greatly help the reader in the Discussion, as a guidance for identifying the CO2 and T effects at a glance.

INTRODUCTION

I suggest shortening the introduction, by reducing general information and focusing directly on the studied pathosystem. For example, the paragraphs describing eCO2 and eT effects (lines 62-78 and 79-91 respectively) should be addressed only to the information already available on the wheat-aphid-BYDV interaction or on close pathosystems.

Line 53-61: the paragraph should be moved and integrated in the last paragraph of the introduction (describing the case-study of the paper) to avoid a break of the description of atmospheric and climatic effects.

Line 58: consider adopting the new “nomenclature” of the types of vector-mediated transmission (ref: Dietzgen et al, 2016: Plant Virus–Insect Vector Interactions: Current and Potential Future Research Directions; Viruses, 8, 303): circulative non-propagative transmission instead of persistent and circulative transmission.

Line 93: “research need to consider changes in different climate and atmospheric variables as a whole”.

Lines 103-108: It is not clear the cause-and-effect relationship of the sentences… Again, are the information essential for the manuscript?

Lines 109-112: these sentences sound redundant (see information in the previous paragraph). 

MATERIALS AND METHODS

Line 161: based on.

Line 195: remove “an individual R. padi was monitored daily”: redundant.

Line 205: I cannot understand the meaning of Md: if it refers to a pre-reproductive period, how the generation of offspring is possible? Please clarify.

Lines 207-209: please clarify the r, Td and RGR formulas: where 0.738 and 0.86 are from?

RESULTS

Try to simplify the text whenever it is possible (i.e. by avoiding comments on non-significant effects and moving statistical values into the figures or figure legends). There are a lot of data presented, and you should help the reader to retain only those which are essential.

Lines 230-1 and throughout the Result section: “There was a statistically significant interaction between CO2&T conditions and virus infection on plant height depending on the week”.

Lines 257-260: replace with: “A decreasing trend in SPAD, which shows changes in chlorophyll content (higher values represent greener plants), was observed since 2 wpi; non-infected plants had higher values of SPAD than infected plants, irrespective of CO2&T conditions (p ≤ 0.05 in all cases, Figure 1c).”

Line 280-283: these details can be retrieved in the corresponding figure, so you can skip them.

Paragraph 3.2: again, skip the details that can be retrieved in the corresponding figures.

Paragraph 3.3: try to be clear about the comparisons you did within each condition vs among conditions; sometimes it is not obvious, for examples the two sentences at lines 338-343 seem to be contradictory.

Table 1. Are all the parameters essential for the paper? Can some analyses be moved to supplemental materials?

DISCUSSION

The discussion should be revised. 1) Start from the obtained results to give possible deductions/conclusions instead of staring from general/available information to adapt the results to them 2) Avoid redundancies and unnecessary information that overshadow the prominent conclusion of this study (which is included in the lines 508-513). Over information are for example in the paragraphs between lines 448-471 and 481-502.

Lines 396-7: “as these two factors are interrelated in the alteration of the climate.”: redundant.

Lines 422-424: the two sentences seem to give the same information: redundant.

Lines 426-431: it is not clear if the eCO2 and eT condition increased or decreased the N content in the host plant: the sentences are inconsistent.

Lines 431-435: it is not clear how the N content (low? High? Specify) can influence aphid fecundity (and then the other aphid life-traits). Apparently, the sentences are lacking in logic.

Line 495: “…no effect on higher trophic level was shown, as R. padi growth rate remained unchanged when aphids were grown on BYDV-PAV infected plants under eCO2&eT compared to aCO2&aT…”.The authors started to face this aspect at line 480: please, write a unique paragraph giving all the conclusions together.

Line 503-507: redundant.

Lines 519: “But this assumption could be only verified if the two factors were analyzed separately.” The authors spent a lot of words to emphasize that this is the first study analysing the combined effects of eCO2&eT, and now they suggest keeping them separately??

CONCLUSIONS

Avoid this chapter and integrate the paragraphs 529-535 and 536-542 in the last part of the discussion (they are partly redundant). Moreover, the authors should clearly indicate the potential effects of their findings on the BYDV disease epidemiology in the future. In my opinion, the aim of the “forecast” studies (like this) must be a contribution in planning an effective management and prevention of the diseases in the next years and not a mere description of biological changes.  

Author Response

Reviewer 2

Comments and Suggestions for Authors

The manuscript “Simultaneous increase in CO2 and temperature alters wheat growth and aphid performance differently 3 depending on virus infection” investigates the combined effect of CO2 and temperature on a plant-insect-virus relationship represented by wheat, Barley yellow dwarf virus (BYDV), and the aphid Rhopalosiphum padi. Wheat traits and aphid behaviour were analysed in presence and absence of virus infection and under two CO2 and T conditions, that represent current and future climatic scenarios. The same trophic interaction was already studied under different CO2 and T conditions separately, and the manuscript provides the first results obtained combining the two abiotic factors. For this reason, this study can provide new information on the possible evolution of the viral disease incidence in a future scenario and is worthy to be considered for publication. However, the manuscript is presented as a narrative (like a thesis) and would be better if presented in a scientific way, clearly pointing out the main outcomes that are relevant for final conclusions. The authors should avoid over information, and turn attention to the impact of their findings on the BYDV disease epidemiology in the future. Due to the great amount of data presented in the Results, I also suggest introducing a concluding figure that visually resumes the significant and non-significant effects of CO2 and T conditions. This would greatly help the reader in the Discussion, as a guidance for identifying the CO2 and T effects at a glance.

As proposed by this reviewer, we have summarized the manuscript and highlighted the main conclusion, moving its position to the beginning of the discussion and including it in the abstract (where it was lacking).

We have eliminated redundant information to be clearer with our results and conclusions.

We have incorporated a conceptual figure to summarize the results.

INTRODUCTION

I suggest shortening the introduction, by reducing general information and focusing directly on the studied pathosystem. For example, the paragraphs describing eCO2 and eT effects (lines 62-78 and 79-91 respectively) should be addressed only to the information already available on the wheat-aphid-BYDV interaction or on close pathosystems.

We have deleted some extra information related with other crop systems.

Line 53-61: the paragraph should be moved and integrated in the last paragraph of the introduction (describing the case-study of the paper) to avoid a break of the description of atmospheric and climatic effects.

We agree with the reviewer and moved the paragraph related with BYDV and R. padi to the end of the introduction, where our pathosystem and research objective are explained.

Line 58: consider adopting the new “nomenclature” of the types of vector-mediated transmission (ref: Dietzgen et al, 2016: Plant Virus–Insect Vector Interactions: Current and Potential Future Research Directions; Viruses, 8, 303): circulative non-propagative transmission instead of persistent and circulative transmission.

We have updated the nomenclature as proposed and included extra references.

Line 93: “research need to consider changes in different climate and atmospheric variables as a whole”.

Sentence rewritten as proposed.

Lines 103-108: It is not clear the cause-and-effect relationship of the sentences… Again, are the information essential for the manuscript?

We have deleted no essential information for our research objective within this paragraph.

Lines 109-112: these sentences sound redundant (see information in the previous paragraph).

We have deleted these sentences as they are similar to the topic of the previous paragraph, and we have done some changes in that paragraph to integrate some deleted information.

MATERIALS AND METHODS

Line 161: based on.

Corrected.

Line 195: remove “an individual R. padi was monitored daily”: redundant.

Done.

Line 205: I cannot understand the meaning of Md: if it refers to a pre-reproductive period, how the generation of offspring is possible? Please clarify.

Regarding this comment and another related comment of Reviewer 1, we have improved the explanation of the aphid biological parameters in Materials & Methods section (L205-214)

Lines 207-209: please clarify the r, Td and RGR formulas: where 0.738 and 0.86 are from?

See previous comment

RESULTS

Try to simplify the text whenever it is possible (i.e. by avoiding comments on non-significant effects and moving statistical values into the figures or figure legends). There are a lot of data presented, and you should help the reader to retain only those which are essential.

We have deleted some non-significant effects.

Lines 230-1 and throughout the Result section: “There was a statistically significant interaction between CO2&T conditions and virus infection on plant height depending on the week”.

Corrected throughout the section.

Lines 257-260: replace with: “A decreasing trend in SPAD, which shows changes in chlorophyll content (higher values represent greener plants), was observed since 2 wpi; non-infected plants had higher values of SPAD than infected plants, irrespective of CO2&T conditions (p ≤ 0.05 in all cases, Figure 1c).”

Done, we have shortened the sentences as proposed.

Line 280-283: these details can be retrieved in the corresponding figure, so you can skip them.

We prefer to maintain the statistical information, as it could be useful and required by other researchers to better appreciate the significant differences.

Paragraph 3.2: again, skip the details that can be retrieved in the corresponding figures.

We have deleted the sentences regarding non-significant differences, as they can be drawn from the corresponding figure.

Paragraph 3.3: try to be clear about the comparisons you did within each condition vs among conditions; sometimes it is not obvious, for examples the two sentences at lines 338-343 seem to be contradictory.

We have deleted the sentence proposed to be clear with the comparison among conditions.

Table 1. Are all the parameters essential for the paper? Can some analyses be moved to supplemental materials?

We would prefer not to remove any biological parameter in Table 1 because there are differences by definition between them, and in order to make our data available and comparable for researchers, as some of them working with rm instead of RGR, or with Tdinstead of d.

DISCUSSION

The discussion should be revised. 1) Start from the obtained results to give possible deductions/conclusions instead of staring from general/available information to adapt the results to them 2) Avoid redundancies and unnecessary information that overshadow the prominent conclusion of this study (which is included in the lines 508-513). Over information are for example in the paragraphs between lines 448-471 and 481-502.

We have move the main conclusion of the study to the beginning of the discussion.

Whenever possible, we have summarized some parts in the discussion. Furthermore, we have deleted duplicated information (that was already in the manuscript) and extra information (about trophic systems in which crops were not cereals).

Lines 396-7: “as these two factors are interrelated in the alteration of the climate.”: redundant.

We agree. Deleted.

Lines 422-424: the two sentences seem to give the same information: redundant.

We agree and have summarized the paragraph.

Lines 426-431: it is not clear if the eCO2 and eT condition increased or decreased the N content in the host plant: the sentences are inconsistent.

We have introduced some changes in the sentence to better explain the statement.

Lines 431-435: it is not clear how the N content (low? High? Specify) can influence aphid fecundity (and then the other aphid life-traits). Apparently, the sentences are lacking in logic.

To clarify this sentence, we have moved the information related to the effect of virus infection on N content to its corresponding position (3 paragraphs below), and we have introduced some referenced information about the relationship between plant nutrition quality/N content and aphid life history parameters.

Line 495: “…no effect on higher trophic level was shown, as R. padi growth rate remained unchanged when aphids were grown on BYDV-PAV infected plants under eCO2&eT compared to aCO2&aT…”.The authors started to face this aspect at line 480: please, write a unique paragraph giving all the conclusions together.

We have deleted some duplications and extra information in the two paragraphs, and finally combined all the information related to the effect of eCO2&eT on virus infection in one paragraph.

Line 503-507: redundant.

We have moved this paragraph to introduce Conclusions section.

Lines 519: “But this assumption could be only verified if the two factors were analyzed separately.” The authors spent a lot of words to emphasize that this is the first study analysing the combined effects of eCO2&eT, and now they suggest keeping them separately??

We agree with the reviewer, our main objective was to work within a more realistic future climate scenario (eCO2&eT), so we have deleted this sentence.

CONCLUSIONS

Avoid this chapter and integrate the paragraphs 529-535 and 536-542 in the last part of the discussion (they are partly redundant). Moreover, the authors should clearly indicate the potential effects of their findings on the BYDV disease epidemiology in the future. In my opinion, the aim of the “forecast” studies (like this) must be a contribution in planning an effective management and prevention of the diseases in the next years and not a mere description of biological changes.  

Conclusions is a compulsory section for Insects, therefore we maintain this section, although we have deleted some redundant information (already exposed in Discussion section), as proposed by the reviewer, and we have added the interesting contributions that s/he have made about the potential of our research in disease epidemiology.

Reviewer 3 Report

Review:  Simultaneous increase in CO2 and temperature alters wheat growth and aphid performance differently depending on virus infection.

The manuscript presents a series of laboratory studies to assess to influence of simultaneous increase in CO2 and temperature combined with virus infection on i) the wheat host-plant growth/performance and ii) aphid vector performance. The studies touch on important aspects of the tripartite plant-insect-pathogen relationship with both theoretical and practical scopes. The interpretation of the results is straightforward. The manuscript is well written and organized.

I identified some minor issues detailed below:

  • While the introduction is exhaustive regarding the effects of elevated CO2 and temperature on host plants and the aphid vector, I find that there is not enough information on the effects of plant viruses. Perhaps you could introduce recent works on the subject and give the general trends observed concerning the Luteovirus (i.e. increase of vector performances most of the time).
  • Perhaps the authors could add some predictions at the end of the introduction to clarify the study question.
  • The authors present a lot of results on both plant and aphid performances/fitness traits. This represents a lot of data and it is a bit difficult to “capture” everything. A conceptual figure or a summary table could facilitate reading.
  • 4a. Can you adjust the y-axis so the difference is easier to visualize? Maybe from 30 to 50%?
  • Table 1. Some parameters seem redundant, to facilitate reading, I would recommend keeping the RM or the RGR but not both which seem identical in terms of variations here (same stats). Likewise, I will only keep the pre-reproductive period or Mean generation time, but not both (again same stats).
  • The Fig 5. is hard to understand and differences are not easy to spot even if the stats indicate strong P < 0.001 differences. I suggest deleting this Figure and keep only Fig. 6 which is easier to read and give the same trend.
  • 398-407. Can you be more specific in your explanations and give the trends (increased/decreased etc…), “influence” is too evasive.
  • 438-447. Can you give more explanations/interpretations to complete the comparisons to the previous studies?
  • 473. Please cite reviews and not only specific studies [77,78]. For example, Eigenbrode et al. 2018 (doi/10.1146/annurev-ento-020117-043119) or Mauck et al. papers (10.1016/bs.aivir.2018.02.007).

Author Response

Reviewer 3

Comments and Suggestions for Authors

Review:  Simultaneous increase in CO2 and temperature alters wheat growth and aphid performance differently depending on virus infection.

The manuscript presents a series of laboratory studies to assess to influence of simultaneous increase in CO2 and temperature combined with virus infection on i) the wheat host-plant growth/performance and ii) aphid vector performance. The studies touch on important aspects of the tripartite plant-insect-pathogen relationship with both theoretical and practical scopes. The interpretation of the results is straightforward. The manuscript is well written and organized.

I identified some minor issues detailed below:

While the introduction is exhaustive regarding the effects of elevated CO2 and temperature on host plants and the aphid vector, I find that there is not enough information on the effects of plant viruses. Perhaps you could introduce recent works on the subject and give the general trends observed concerning the Luteovirus (i.e. increase of vector performances most of the time).

Perhaps the authors could add some predictions at the end of the introduction to clarify the study question.

The authors present a lot of results on both plant and aphid performances/fitness traits. This represents a lot of data and it is a bit difficult to “capture” everything. A conceptual figure or a summary table could facilitate reading.

The information about the Luteovirus has been moved to the penultimate paragraph and the effects of eCO2 on virus vector are indicated in L78-80.

We have incorporate a conceptual figure to summarize the results.

4a. Can you adjust the y-axis so the difference is easier to visualize? Maybe from 30 to 50%?

In order to be consistent with the other two charts in this figure, we would prefer to maintain Y-axis starting from 0. Furthermore, the statistically significant difference between climate conditions could be appreciate maintaining the Y-axis as it is.

Table 1. Some parameters seem redundant, to facilitate reading, I would recommend keeping the RM or the RGR but not both which seem identical in terms of variations here (same stats). Likewise, I will only keep the pre-reproductive period or Mean generation time, but not both (again same stats).

We would prefer not to remove any biological parameter because there are differences by definition between them, and in order to make our data available and comparable for researchers, as some of them working with rm instead of RGR, or with Td instead of d.

The Fig 5. is hard to understand and differences are not easy to spot even if the stats indicate strong P < 0.001 differences. I suggest deleting this Figure and keep only Fig. 6 which is easier to read and give the same trend.

We would prefer to maintain the two figures. In Figure 5, the reader can observe the daily trend in fecundity, which rises rapidly the first days and drops from day 3 onwards, commonly shown in aphid fecundity graphs, and could check the fecundity in a specific day. In Figure 6, the reader observes the total fecundity (i.e. the sum of the daily fecundity) which provides an overall information.

398-407. Can you be more specific in your explanations and give the trends (increased/decreased etc…), “influence” is too evasive.

We have introduced some changes in the explanation about the effect of virus infection and CO2&T conditions in plant parameters to be more exact.

438-447. Can you give more explanations/interpretations to complete the comparisons to the previous studies?

We have included the interpretation of our results and different trends in population dynamics in the future, that it was lacking.

  1. Please cite reviews and not only specific studies [77,78]. For example, Eigenbrode et al. 2018 (doi/10.1146/annurev-ento-020117-043119) or Mauck et al. papers (10.1016/bs.aivir.2018.02.007).

We appreciate the suggestion and incorporate the reviews to better confirm the statement.